# Giant bulk photovoltaic effect driven by the wall-to-wall charge shift in WS$_2$ nanotubes

Bumseop Kim[1], Noejung Park [1✉] & Jeongwoo Kim [2✉]

The intrinsic light–matter characteristics of transition-metal dichalcogenides have not only been of great scientific interest but have also provided novel opportunities for the development of advanced optoelectronic devices. Among the family of transition-metal dichalcogenide structures, the one-dimensional nanotube is particularly attractive because it produces a spontaneous photocurrent that is prohibited in its higher-dimensional counterparts. Here, we show that WS$_2$ nanotubes exhibit a giant shift current near the infrared region, amounting to four times the previously reported values in the higher frequency range. The wall-to-wall charge shift constitutes a key advantage of the one-dimensional nanotube geometry, and we consider a Janus-type heteroatomic configuration that can maximize this interwall effect. To assess the nonlinear effect of a strong field and the nonadiabatic effect of atomic motion, we carried out direct real-time integration of the photoinduced current using time-dependent density functional theory. Our findings provide a solid basis for a complete quantum mechanical understanding of the unique light–matter interaction hidden in the geometric characteristics of the reduced dimension.

[1] Department of Physics, Ulsan National Institute of Science and Technology, Ulsan 689-798, Korea. [2] Department of Physics, Incheon National University, Incheon 406-772, Korea. ✉email: noejung@unist.ac.kr; kjwlou@inu.ac.kr

Atomically thin transition-metal dichalcogenides (TMDs), which consist of transition-metal atoms sandwiched between layers of chalcogen atoms, have attracted extensive interest as a novel platform hosting various quantum phenomena[1–4]. Single-layer TMDs exhibit a direct bandgap with a unique spin–valley coupling induced by the broken inversion symmetry[5–7], and the less-screened Coulomb interaction gives rise to the strong excitonic transitions including higher-order excitonic states[8–12]. In addition, large photoluminescence[13,14], fast photocurrent switching[15–17], and high photoresponsivity[18–20] have been demonstrated and various advanced optoelectronic applications (e.g., photodetectors[21–23], photovoltaic devices[24,25], and light-emitting diodes[26,27]) have been envisioned for van der Waals (vdW)-stacked TMD layers.

Despite the various intriguing light–matter characteristics of TMD layers, the spontaneous photocurrent without external bias, which has been referred to as the bulk photovoltaic effect (BPVE), is absent because of the inherent mirror symmetries. Intensive efforts have been devoted to unleashing the intrinsic optoelectronic advantages confined under the symmetry constraints of TMDs, such as by using electrostatic gating[25,28], an in-plane/out-of-plane $p$–$n$ junction[29], the broken symmetry of an edge[30,31], and vdW heterostacks[32–34]. Moreover, an innovative approach to the TMD-based BPVE was initiated by a recent study of one-dimensional (1D) WS$_2$ nanotubes[35]. The obtained short-circuit current of a WS$_2$ nanotube was found to be a few orders of magnitude larger than that of conventional photovoltaic materials. This dimensionality reduction, from a two-dimensional (2D) layer to a 1D nanotube, is now attracting extensive attention as a breakthrough that overcomes the limitations of conventional devices based on junctions or heterointerfaces of materials[36,37].

Nevertheless, the physical mechanism underlying the large BPVE in TMD nanotubes has not yet been elucidated. A spontaneous photocurrent (shift current) is induced by the spatial charge shift between occupied and unoccupied states via optical excitation of polar materials[38]. As atomic structures, electronic band dispersions, and electric polarizations are closely intertwined in this phenomenon[39,40], the fundamental question is whether any particular geometrical advantages are inherent to the 1D structure[35]. In these inspections, the justification of the perturbation-based theories and the consideration of the nonlinear effect of strong fields also need to be addressed to achieve complete understanding of the behavior of TMD nanotubes under practical application conditions.

In the present work, we address the aforementioned questions through second-order perturbation analysis[41] based on static density functional theory (DFT), together with real-time calculations using time-dependent density functional theory (TDDFT). We find that the BPVE characteristics of nanotubes can be decomposed into intra-tube and inter-tube effects. In WS$_2$ single-walled nanotubes (SWNTs), the shift current is dominated by the intra-tube effect triggered by $d$–$d$ transitions of W atoms[42], leading to a shift current three times larger than that of photovoltaic perovskites[43]. For WS$_2$ double-walled nanotubes (DWNTs), the wall-to-wall charge shift becomes substantial, which leads to a shift current near the infrared region four times greater than that previously reported[35]. By carrying out real-time integration of the current using real-time time-dependent density functional theory (rt-TDDFT) calculations, we find that the nonlinear effect of a strong field and the nonadiabatic effects of atomic motions can be critically effective for generating photocurrent. We propose that our results can be used as a guiding principle for designing advanced optoelectronic applications in reduced dimensions.

## Results

**Atomic structure and spontaneous polarization of SWNTs.** As electric polarization is an essential ingredient for the BPVE, we investigate the electric polarization of SWNTs with various diameters. The geometry of the nanotubes is determined by the rolling-up direction, which can be characterized by the so-called chiral vector[44,45], which is given as $\vec{C} = n\vec{a}_1 + m\vec{a}_2 \equiv (n, m)$, where $\vec{a}_1$ and $\vec{a}_2$ are the unit vectors of the hexagonal 2H phase of WS$_2$ and $n$ and $m$ are integers. In the present study, we consider zigzag, armchair, and chiral SWNTs, corresponding to $m = 0$, $m = n$, and $m \neq n$, respectively.

As illustrated in Fig. 1a, mirror symmetry ($M_{xy}$), which is inherited from two-dimensional WS$_2$, is present in the armchair SWNTs, removing the electric polarization (Fig. 1c)[35,46,47]. On the other hand, the mirror symmetry is broken for the zigzag SWNTs (Fig. 1b and Supplementary Fig. 1), leading to spontaneous polarization along the $z$-axis (Fig. 1d). When we assume the close-pack triangular lattice of nanotube bundles, the estimated polarization density (0.82–1.52 C/m$^2$) of the zigzag SWNT is comparable to or even greater than that of representative polar materials such as BaTiO$_3$ (0.26 C/m$^2$) and BiFeO$_3$ (0.9 C/m$^2$)[48,49] (see Supplementary note 1). The main contributor to this large electric polarization in the axial direction is the ionic part, and the polarization normalized by the tube diameter (the polarization divided by $n$) approaches the polarization density of a TMD monolayer (Supplementary Fig. 2)[50–52]. The electric polarization of chiral SWNTs is

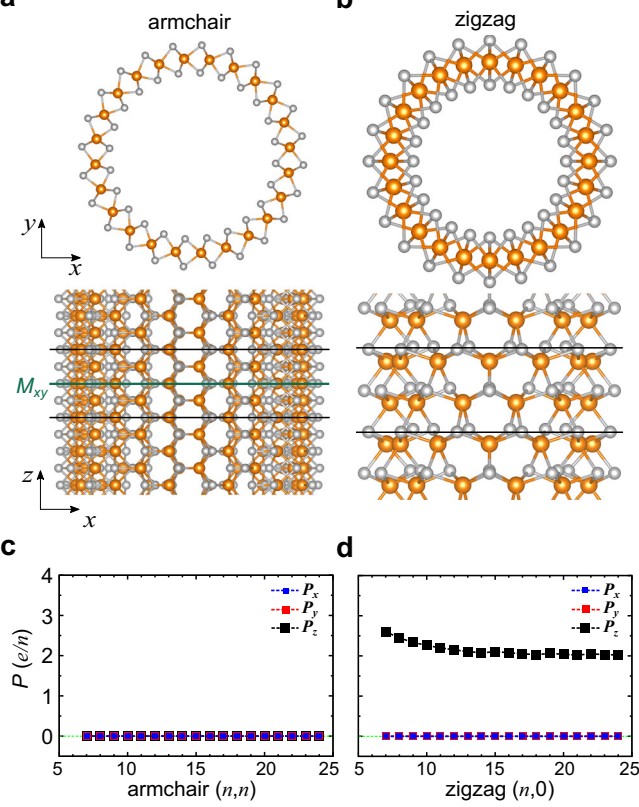

**Fig. 1 Atomic structure and electric polarization of TMD SWNTs.**
**a–b** Atomic illustration of armchair and zigzag SWNTs. The armchair SWNT is mirror symmetric ($M_{xy}$), whereas the zigzag one is not. The orange and gray spheres represent W and S atoms, respectively.
**c–d** Variation of the electric polarization of armchair and zigzag SWNTs with various diameters.

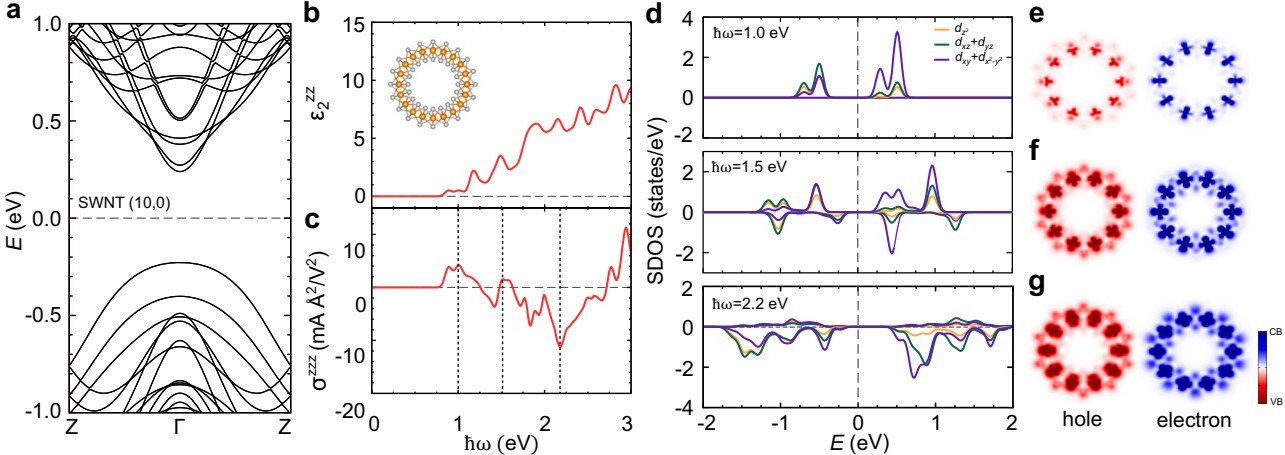

**Fig. 2 Electronic and optoelectronic properties of the zigzag (10,0) SWNT. a** Calculated band structure of the SWNT. **b** Imaginary parts of the dielectric constants of the SWNT. The inset indicates the cross-section of the zigzag SWNT. **c** Calculated shift-current spectrum of the SWNT with respect to the frequency of the applied light. **d** The shift-current-weighted density of states (SDOS) of the SWNT responsible for the shift-current peaks indicated by the black dotted lines ($\hbar\omega = 1.0, 1.5, 2.2$ eV) in (**c**). **e–g** Real-space representation of the hole and electron carrier densities of (**e**) $\hbar\omega = 1.0$ eV, (**f**) $\hbar\omega = 1.5$ eV, and (**g**) $\hbar\omega = 2.2$ eV excitations. The carrier density is proportional to the optical absorption of (**b**).

intermediate between those of the zigzag and armchair SWNTs, depending on the rolling-up direction (Supplementary Fig. 3). Although various chiral nanotubes have been speculated to be coexist in the condensed phases of actual synthesized samples[35], the zigzag nanotube is likely to be the greatest contributor to the photovoltaic property because of its apparent axial polarization. Hereafter, we focus on the electronic and optoelectronic properties of the zigzag nanotube to elucidate the fundamental origin of the experimentally observed large BPVE in WS$_2$ nanotubes.

**Electronic and optoelectronic properties of the zigzag SWNT.**
To identify the characteristics of the electronic structure of the zigzag SWNTs relevant to the shift-current generation, we calculate its band structures and shift-current spectra. As presented in Fig. 2a, the zigzag (10,0) SWNT has a direct bandgap at the Brillouin zone center (Γ point)[42,53]. The band structures of other zigzag SWNTs have similar overall features, and the size of the bandgap increases with the tube diameter (Supplementary Fig. 4) in good agreement with previous results[50,51]. As the bandgap increases with the tube size, the onset frequency slightly shifts up and the absorption strength marginally decreases[54]. Nevertheless, the overall shape of the absorption spectrum is maintained irrespective of the tube size (Fig. 2b and Supplementary Fig. 4)[47]. The band-edge excitation is almost negligible, and dominant absorption peaks are observed in the higher-energy region above 1.0 eV (Fig. 2b). On the basis of the electronic and optical properties, we evaluate the shift-current spectra (i.e., the response function between the shift-current density and external electric fields) using second-order perturbation theory[41] (Fig. 2c). The spectrum of the zigzag (10,0) SWNT shows large shift-current peaks: 6.2 mA Å$^2$/V$^2$ at $\hbar\omega = 1.0$ eV and $-14.2$ mA Å$^2$/V$^2$ at 2.2 eV. When the SWNTs are assumed to form a bundle closely packed into the 2D hexagonal lattice, these shift-current values correspond to 22.1 and 50.6 μA/V$^2$, respectively, which are much larger than the maximum values of the known photovoltaic perovskite oxides, such as PbTiO$_3$ (16.9 μA/V$^2$) and BaTiO$_3$ (16.1 μA/V$^2$), near the visible-light region[44]. The shift current spectra of various (n,0) SWNTs are presented in Supplementary Fig. 4 in the supplementary data. The overall size of the shift current somewhat decreases with increasing tube size, which

depends on the optical absorption associated with the bandgap of the SWNTs[51,55]. (Supplementary Fig. 4)

To elucidate the electronic origin of the enhanced shift current, we compute the shift-current-weighted density of states (SDOS), which is defined by the orbital-projected density of states (Supplementary Fig. 5) weighted by the contribution of each state to the shift current (Fig. 2d)[40]. The orbital character of hole (electron) carriers contributing to the shift current is represented by the spectral density below (above) the Fermi level in the SDOS of the zigzag (10,0) SWNT. We investigate the transitions of three different energy regions ($\hbar\omega = 1.0, 1.5, 2.2$ eV) corresponding the vertical dotted lines in Fig. 2c. We find that, rather than the $p$–$d$ or $p$–$p$ transitions, the transitions between different $d$ orbitals are dominant in the generation of the shift current, irrespective of the light frequency (Supplementary Fig. 6)[53]. In all these transitions, a substantial amount of the axial orbital ($d_{z^2}$) of the initial state is excited into another $d$ orbital. For the low-energy transition ($\hbar\omega = 1.0$ eV), the excited electron states mainly consist of the planar orbital ($d_{x^2-y^2} + d_{xy}$), whereas the hole states are derived from a mixture of various $d$ orbitals (Fig. 2e). The positive SDOS indicates that the shift current flows in only one direction (the top panel in Fig. 2d). With increasing excitation frequency ($\hbar\omega = 1.5$ eV), a small negative SDOS with a large planar orbital contribution begins to appear above the Fermi level (the middle panel in Fig. 2d), which contributes to the shift currents developing in the opposite direction. As a result of the cancellation, the shift-current peak becomes smaller near $\hbar\omega = 1.5$ eV (Fig. 2c). Under the higher-frequency light radiation ($\hbar\omega = 2.2$ eV), the positive SDOS is strongly suppressed, leading to a large negative shift-current peak. On the basis of the shift current and SDOS result, we can control the direction and the magnitude of the shift current in SWNTs by tuning the applied light.

To examine possible artifacts coming from the approximated density functions, we carry out the same calculations with the hybrid functional and confirm that essentially the same electronic and optoelectronic properties of the zigzag SWNTs are obtained (Supplementary Fig. 7). We also confirmed that various other TMD nanotubes result in qualitatively the same shift current characteristics (Supplementary Fig. 8).

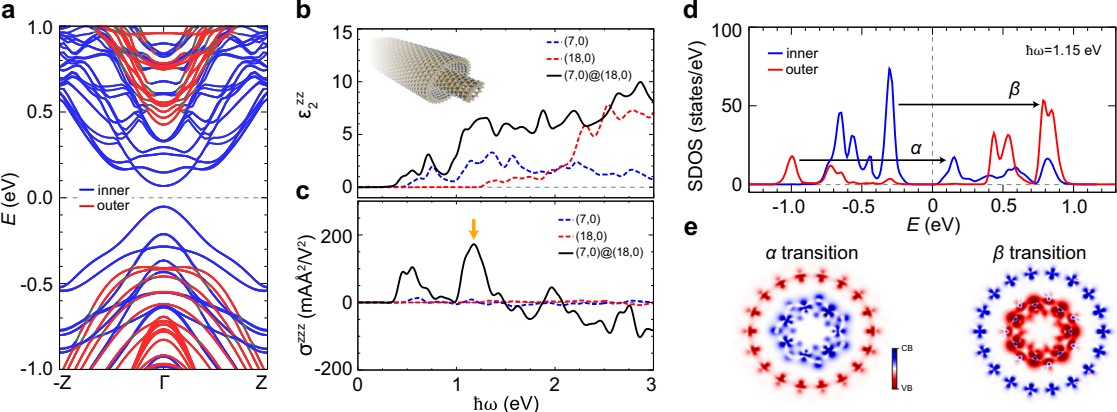

**Fig. 3 Electronic and optoelectronic properties of the (7,0)@(18,0) DWNT. a** Calculated band structure of the DWNT. The states from the inner (outer) tube are represented by blue (red) lines. **b** Imaginary parts of the dielectric constants of the individual (7,0) and (18,0) SWNTs, and the DWNT. **c** Calculated shift-current spectra of the individual (7,0) and the (18,0) SWNTs and the DWNT with respect to the frequency of the applied light. **d** Shift-current-weighted density of states (SDOS) of the DWNT corresponding to the shift-current peak ($\hbar\omega = 1.15$ eV) denoted by the orange arrow in (**c**). Two interwall transitions are labeled as $\alpha$ and $\beta$ transitions. The states from the inner (outer) tube are represented by blue (red) lines. **e** Real-space representation of the hole (red) and electron (blue) carrier density of the $\alpha$ and $\beta$ transitions.

**Electronic and optoelectronic properties of the zigzag DWNT.** The results discussed above clearly indicate that the optoelectronic nature of TMD materials can be activated by the dimensionality reduction to the 1D nanotube structure. Here, we extend the study to multiwall structures, with particular focus on the interwall effect, given that the samples synthesized in the experiment are thought to be mostly multiwalled concentric tubes[35]. To investigate whether the interwall interaction gives rise to an additional enhancement beyond the mere sum of individual tubes, we selected the (7,0)@(18,0) DWNT as a minimal example of multiwalled tubes. As shown in Fig. 3a, the bandgap of the zigzag (7,0)@(18,0) DWNT (0.18 eV) is slightly smaller than that of the inner SWNT (0.21 eV), leading to a marginal downshift of the absorption peak (Fig. 3b). Overall, the absorption intensity of the DWNT is comparable to the summation of the absorption intensities of the individual SWNTs (Fig. 3b). However, the DWNT exhibits an order-of-magnitude larger shift current compared with that of the constituent SWNTs (Fig. 3c). As indicated by the downward arrow in Fig. 3c, the low-frequency peak value (193 mA Å$^2$/V$^2$) is almost 20 times larger than that of the SWNTs (5.9–16.6 mA Å$^2$/V$^2$, Fig. 2c). This giant shift-current peak indicates that the DWNT possesses a new photoactive conduction channel that is not inherent to the individual SWNTs. This enhancement owing to the inter-wall effect is persistent even for the large wall-to-wall distance. In the supplementary data, Supplementary Fig. 9, we show that the inter-wall effect decays with the wall-to-wall distance, but still remains until the distance reaches 11 Å.

To elucidate the effect of the wall-to-wall transitions more explicitly, as depicted in Fig. 3d, we analyze the origin of the large shift-current peak using the SDOS projected into $d$ orbitals of each inner and outer tube. The spatial distributions of the hole and electron carrier densities associated with the two distinct interwall excitations, denoted as $\alpha$ and $\beta$ in Fig. 3d, are presented in Fig. 3e, which obviously confirms that each carrier belongs to each individual tube. This result clearly indicates that newly emergent interwall excitation channels give rise to the giant BPVE. Notably, the interwall charge-shifting mechanism is not necessarily confined to the coaxial nanotubes. We verify the appearance of the interwall effect in a SWNT bundle (Supplementary Fig. 10) and an MoS$_2$ DWNT (another member of the TMD nanotube family) (Supplementary Fig. 11). Our results imply that the TMD multiwalled nanotubes, which are equipped

with interwall charge-shifting channels, have excellent potential for low-frequency optoelectronic applications. Compared with conventional solar-energy-harvesting devices, which are centered around the visible or higher-frequency regime,[35] the TMD nanotubes have an advantage in the lower-frequency region.

The interwall charge-shifting mechanism, shown to yield the dramatic increase in the shift current, is not limited to the DWNTs but extensively applicable to general multi-walled nanotubes[35]. The effect in a thick multi-walled tube can be estimated from the comparison between the DWNT and the triple-walled nanotube (TWNT), as summarized in Supplementary Fig. 12. While the DWNT presents apparent enhancements over the SWNT, the shift current and absorption strength of the TWNT is largely comparable to that of DWNT in the low-energy region (<2 eV); the effect of triple walls is appreciable only in the higher-energy region (> 2 eV). The larger tube, in the outer shell, possess a larger bandgap, and the interwall mechanisms between the pairs of outer walls are manifested in higher-energy region. This indicates that, even in thick multi-walled tubes, the interwall effect is mostly governed by the same effect in the DWNT. Furthermore, this intriguing advantage of the interwall charge shift is distinct from the effect of dimensionality lowering, as it cannot be obtained from the monolayers; the effect of dimensional crossover, from 2D layers to 1D nanotubes, was highlighted in the discussion in ref. [35]. For a side discussion, condensed aggregates of realistic samples certainly include portions of chiral nanotubes. As we present in Supplementary Fig. 9, the interwall charge-shifting mechanism is quite persistent over large wall-to-wall separations, and the main optical absorption responsible for the large shift current is not driven by the band-edge configurations but by the interwall effect between the $d$ orbitals states of neighboring tubes. Thus, the interwall effect we discussed for the zigzag DWNTs could be extended to the multi-walled TMDC nanotubes with various chiralities.

**Photovoltaic effect of Janus-type WSSe nanotubes.** The important contribution of the interwall charge shift in multiwalled TMD nanotubes, as discussed in the preceding section, inspired us to consider whether Janus-type multiwalled tubes could lead to a further enhancement of the inter-tube charge-shifting mechanism because of their built-in radial electric

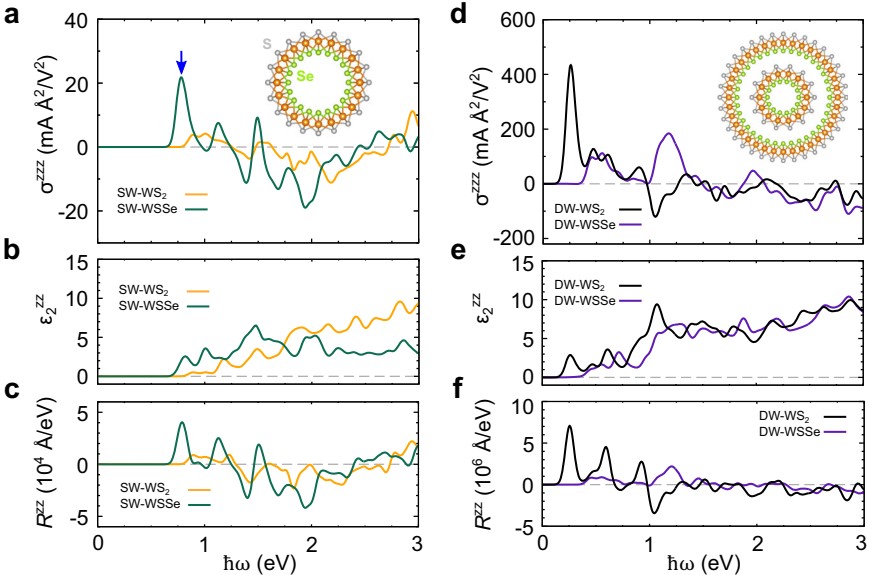

**Fig. 4 The photovoltaic effect of Janus-type WSSe nanotubes in comparison with that of WS₂ nanotubes. a**–**c a** Calculated shift-current spectra, **b** imaginary parts of the dielectric constants, and **c** shift vector of the (10,0) WS₂ SWNT (SW-WS₂) and a (10,0) WSSe SWNT (SW-WSSe) with respect to the frequency of the applied light. **d**–**f d** Calculated shift-current spectra, **e** imaginary parts of the dielectric constants, and **f** shift vector of the (7,0) @(18,0) WS₂ DWNT (DW-WS₂) and (7,0)@(18,0) WSSe DWNT (DW-WSSe) with respect to the frequency of the applied light.

field[56,57]. The Janus-type nanotube, whose inner and outer chalcogen layers are composed of S and Se atoms[58], respectively, is illustrated in the inset of Fig. 4a and is hereafter referred to as WSSe nanotube. We first examine the optoelectronic properties of the (10,0) SWNT of the WSSe. Although the overall shift-current spectrum exhibits similar features as that of the WS₂ (10,0) SWNT, the spectrum of the Janus-type SWNT has a prominent low-energy peak, as indicated by the blue arrow in Fig. 4a. This large shift-current peak is attributable to the increased carrier population (optical absorption: Fig. 4b) and the larger shift vector (Fig. 4c). Shift vector is the charge-center difference between initial and final states on the excitation[41]. The SDOS of the (10,0) WSSe SWNT confirms that the low-energy excitation is dominated by the *d*–*d* transition (Supplementary Fig. 13); various other combinations of the Janus-type TMD nanotubes exhibit similar trends (Supplementary Fig. 14).

For the case of the Janus-type coaxial DWNT, as expected, the shift current is dramatically enhanced, particularly in the low-energy region (Fig. 4d). Whereas the optical absorption does not substantially differ from that of the WS₂ DWNT (Fig. 4e), the value of the shift vector for the Janus DWNTs is more than three times greater than that for the pristine DWNTs (Fig. 4f). The large shift vector of the Janus DWNT is obviously ascribed to the radial built-in potential due to the compositional asymmetry between the interfacing walls. Various other compositions of the Janus-type DWNTs are also considered, which exhibit similar features, as summarized in Supplementary Fig. 15.

**Real-time ab initio study of the charge-shifting dynamics: the nonlinear effect of a strong field and the nonadiabatic effect of atomic motions.** The shift-current results presented in the preceding section are all based on perturbation theory. To extend our understanding of the shift-current generation mechanism beyond the limitations of the linear response theories, we here investigate the time-dependent photocurrent using rt-TDDFT calculations, focusing on the effects of a strong field and atomic motion. The details of our rt-TDDFT calculation and the definition of the

current-related quantities, including all even-order responses ($J_{even}(t)$), are summarized in the Methods section. As predicted by our shift-current calculation (Fig. 5a), the zigzag (7,0) SWNT exhibits distinct responses depending on the frequency of the light ($\hbar\omega = 0.9$ and 1.7 eV) for a given intensity ($6.05 \times 10^{10}$ W/cm²). In this scheme, the time-dependent carrier population is obtained through the projection of the time-evolving wavefunctions onto the ground states (Fig. 5b). For both frequencies, the electron/hole-carrier density of states monotonically increases with time, implying a charge shift through carrier excitation, in agreement with the second-order perturbation theories discussed above. To estimate the direct constant current unambiguously, as presented in Fig. 5, the even-order real-time currents are employed ($J_{even}(t) = \frac{(J_+(t)+J_-(t))}{2}$). In this computation, we focus on the linearly polarized light and the injection current is not reflected in the time profile of the current[41]. When the light with frequency $\hbar\omega = 0.9$ eV is applied, the carrier excitation occurs mainly near the $\Gamma$ point in second-order perturbation theory (Supplementary Fig. 16), leading to the two pronounced peaks in Fig. 5b. However, when the light frequency is $\hbar\omega = 1.7$ eV, the absorption arises over a wide range of the momentum space (Supplementary Fig. 16), enabling substantial spread of electron/hole carriers beyond the $\Gamma$ point (Fig. 5b). The consistency between the perturbation results and the direct real-time integration of the photocurrent using the rt-TDDFT approach is verified in this section. As the external fields are implemented into the vector potential of the density functional Hamiltonian, the latter method can naturally be applied in the case of a strong field; we focus on this approach in the following paragraphs.

Here, we consider the effect of field strength on the generation of the photocurrent. For a more comprehensive comparison, we calculate the normalized time-averaged currents, defined as $\widetilde{J}_{avg}(t) = J_{avg}(t)/I$, where $I$ is the intensity of the external electric field. For the given field ($\hbar\omega = 0.9$ eV), the normalized photocurrent of the zigzag (7,0) SWNT is presented in Fig. 5c. Interestingly, the current direction is reversed when the field strength is increased beyond $I \geq 10^{12}$ W/cm². To analyze this

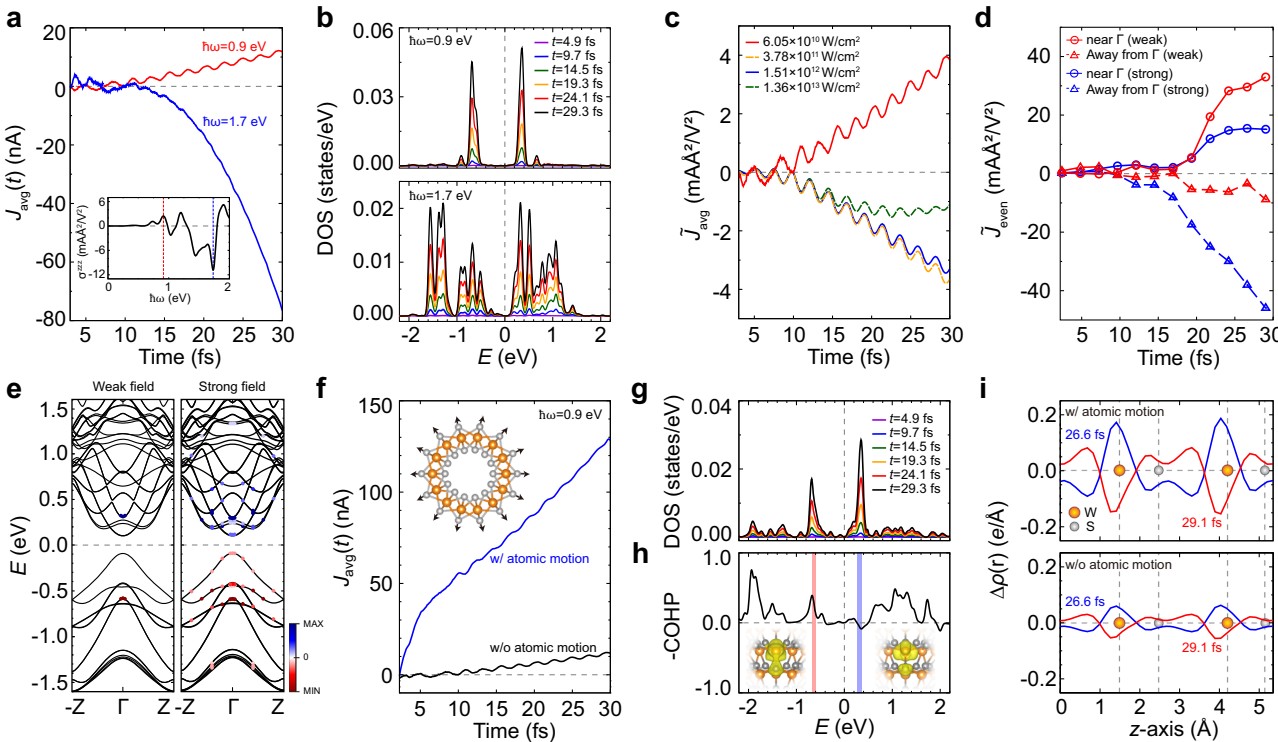

**Fig. 5 The effect of strong field and atomic motion on the photocurrent of the zigzag (7,0) WS$_2$ SWNT. a** Calculated time-averaged second-order currents created by external fields with an intensity of $6.05 \times 10^{10}$ W/cm$^2$ and frequencies of $\hbar\omega = 0.9$ and 1.7 eV. The quantitative definition of $J_{avg}$ is described in the main text. The inset shows the shift-current spectra obtained by the same scheme as the spectra in Fig. 2c. Red and blue dotted vertical lines in the inset indicate $\hbar\omega = 0.9$ and 1.7 eV, respectively. **b** Time evolution of the electron/hole-carrier density of states of the SWNT excited by the light frequencies of $\hbar\omega = 0.9$ and 1.7 eV. The Fermi level is set to zero. **c** The time-averaged currents normalized by the light intensity ($I$) for the frequency of $\hbar\omega = 0.9$ eV, which is defined as $\tilde{J}_{avg} = J_{avg}/I$ . **d** The real-time second-order currents normalized by the field intensity, defined as $\tilde{J}_{even}(t) = J_{even}/I$ for the weak ($I = 6.05 \times 10^{10}$ W/cm$^2$) and strong ($I = 1.51 \times 10^{12}$ W/cm$^2$) fields. Here, the contributions are decomposed into that of the zone-center states near $\Gamma$ ($k < 0.007$ Å$^{-1}$) and the other states ($k > 0.007$ Å$^{-1}$) states. **e** Momentum-resolved electron/hole-carrier of the weak ($I = 6.05 \times 10^{10}$ W/cm$^2$) and strong ($I = 1.51 \times 10^{12}$ W/cm$^2$) fields. The maxima of the electron and hole-carrier (MAX and MIN) are represented by dark-blue and dark-red spheres, respectively. **f** Calculated time-averaged second-order currents ($J_{avg}$) created by an external field ($\hbar\omega = 0.9$ eV) with/without atomic motion. The inset is a schematic of the atomic motion of the SWNT caused by the carrier excitation. **g** Time evolution of the electron/hole-carrier density of states of W atoms, excited by the external field ($\hbar\omega = 0.9$ eV), with the atomic motions of the Ehrenfest dynamics. The Fermi level is set to zero. **h** Crystal orbital Hamilton populations (COHPs) of the (7,0) SWNT, characterizing the bonding (+ sign) and antibonding nature (− sign) of the wavefunctions. The initial (bonding) and final (antibonding) states on the transitions are denoted by red and blue vertical lines, respectively. The inset of the partial charge density confirms the bonding and antibonding characters of the initial and final states. **i** The plane summation of the charge density difference between the excited and the ground states of the SWNT with/without atomic motions.

effect of field strength, we compare the contribution from the zone-center states near the Γ point ($k < 0.007$ Å$^{-1}$) and other states of the Brillouin zone ($k > 0.007$ Å$^{-1}$), as summarized in Fig. 5d. For the weak field ($I = 6.05 \times 10^{10}$ W/cm$^2$), as shown in Fig. 5e, the photocurrent is dominated by the zone-center state. However, under the strong field ($I = 1.51 \times 10^{12}$ W/cm$^2$), the contribution of the non-Γ region becomes substantial and the current direction is flipped. By projecting the time-evolving Kohn–Sham states into the given ground state, we traced the band-resolved time-dependent carrier populations (Fig. 5e). The band-resolved carriers are depicted by colored spheres, whose scale is normalized by the maximum value of the weak-field case within the time window of $0 \leq t \leq 30$ fs. The strong field enables the carrier excitations to be spread over non-Γ regions through a virtual multilevel transition, resulting in the flipping of the photocurrent direction (Fig. 5c, d).

To examine the effect of atomic motions, accompanied by the carrier excitation on the shift-current generation mechanism, we repeat the same real-time photocurrent calculations through the Ehrenfest dynamics, in which atomic motions are progressed through the instantaneous forces[59]. We tested with the light

frequencies corresponding to the peaks in the shift current spectra (the inset of Fig. 5a), and hereafter, we mainly present the results with the frequency of $\hbar\omega = 0.9$ eV. The electronic excitation causes the spontaneous displacement of outer sulfur atoms along the radial direction (inset of Fig. 5f), which greatly enhances the generated photocurrent. To understand the large enhancement of the photocurrent induced by the atomic motion, we examine the time evolution of the projected density of states for W atoms. When the atoms are fixed in the ground-state configuration, the states near the Fermi level, which are mostly attributable to W orbitals, are primarily responsible for the carrier excitation (Fig. 5b and Supplementary Fig. 17). The overall time-evolving characters of the main electron/hole peaks are preserved, even in the presence of Ehrenfest atomic movements; however, the carriers are dispersed over a wider energy range (Fig. 5g). The main electron and hole peaks are located at the local maximum and minimum points of the crystal orbital Hamilton populations (COHPs), as denoted by the vertical red and blue lines, respectively, in Fig. 5h[60]. Notably, the positive and negative values in the plot of the COHP correspond to the bonding and antibonding character, respectively. The aforementioned

excitations weaken the bond strength by increasing antibonding character (Fig. 5h), which triggers the outward displacement of the S atoms. The atomic motion promotes orbital mixing, resulting in a wider dispersion of the electron/hole carriers (Fig. 5b, g). This increased antibonding character induces segregation of the outer S atoms, which further promotes the charge transfer to the W and the inner S atoms (Supplementary Fig. 18). To visualize this redistribution of the excited carrier, we plot the variation of the instantaneous charge density from the ground state in Fig. 5i. The number of excited carriers is substantially increased by the elongation of the W–S distance, and they oscillate back and forth in synchronization with the frequency of the external field. As the carriers are more loosely bound to the W atoms, they become more vulnerable to the shifting in response to the field, resulting in the large photocurrent observed in Fig. 5f. We repeated the same calculations with the geometries fixed at several instantaneous positions; the enhancement is negligible, and the obtained photocurrents are all similar to that of the equilibrium geometry (Supplementary Fig. 19). This Ehrenfest dynamics represents a substantial nonadiabatic effect of atomic motions on the photocurrent. We also carry out the same calculations with $\hbar\omega = 1.7$ eV and find a similar enhancement of the photocurrent (Supplementary Fig. 20).

## Discussion

In summary, we examined the underlying physics of the giant BPVE of TMD nanotubes by using perturbation theory analysis, together with ab initio real-time simulations of the photocurrent. We found that the unique interwall charge shift of the multi-walled coaxial tubes is the primary factor responsible for the large BPVE of the nanotubes. We predicted that the shift current near the infrared region can be four times larger than the maximum value known in the high-frequency region. As an example structure that possesses a strong intrinsic advantage in the wall-to-wall charge shift, we considered Janus-type nanotubes. Beyond the limitations of perturbation methods, we addressed the nonlinear effect of strong fields and the nonadiabatic effect induced by atomic motion by performing rt-TDDFT calculations. We found that the direction and the magnitude of the photocurrent could be controlled by tuning the field frequency and intensity. Through simulations of the Ehrenfest dynamics, we found that the electronic high excitations were relaxed into the radial breathing motions of atoms, which constitutes an additional unique advantage of the nanotube geometry. Our results suggest that the unique geometrical advantage of TMD nanotubes is not simply caused by the intrinsic symmetry lowering of their 1D structure but is mainly attributable to the unique wall-to-wall charge shift combined with the nonadiabatic effect of atomic relaxation.

## Methods

**Electronic structure calculation**. We performed density functional theory (DFT) calculations via the projector-augmented plane-wave method[61], as implemented in the Vienna Ab initio Simulation Package (VASP)[62]. The Perdew–Burke–Ernzerhof (PBE) functional of the generalized gradient approximation[63] was used to describe the exchange–correlation interactions among electrons. We also used the Heyd–Scuseria–Ernzerhof (HSE) hybrid functional[64] to cross-check the calculation of the PBE functional. The spin–orbit coupling effect was included in our electronic structure calculations. Atomic relaxations that included the effect of the vdW interactions were carried out until the maximum forces were less than 0.001 eV Å$^{-1}$. The vacuum layer was set to >15 Å to simulate isolated WS$_2$ nanotubes. The energy cutoff for the plane-wave-basis expansion was selected to be 500 eV. We used a $1 \times 1 \times 10$ $k$-point grid for single-walled WS$_2$ nanotubes and a $1 \times 1 \times 5$ $k$-point grid for double-walled WS$_2$ nanotubes. The electric polarization of the nanotubes was estimated using the Berry phase method[65]

**Second-order optical response using perturbation theory**. The shift-current spectra were evaluated from the tight-binding Hamiltonian based on the maximally

localized Wannier functions[66] using the second-order optical response formalism[39–41]. For the shift-current estimation, a $3 \times 3 \times 100$ $k$-point grid was used in our calculations, which provide a sufficiently dense mesh, leading to the well converged value of the shift-current spectra. To remove the ambiguity in the presentation of the dielectric constant of nanotube caused by the vacuum region in the plane direction, we normalize the optical/optoelectrical quantities by the unit length in the axial direction instead of the volume of a supercell[54,67]. In this regard, the shift current spectra for the 1D nanotube, in the present work, is presented in the unit of A Å$^2$/V$^2$. The calculation of the shift currents, in the scope of the present work, does not include excitonic effects. Since the excitonic effects usually dominate in the absorption peaks near band-edge transitions[8,9], our analysis on the shift current generation mechanism in higher-energy regions, particularly for the interwall charge shift mechanism—the main focus of the present study—is still valid. Note that previous theoretical works without considering the excitonic effects also successfully reproduced the experimental results[43].

**Direct real-time calculation of the current $J(t)$ using the rt-TDDFT**. To examine the electric current of the excited state, we performed rt-TDDFT calculations using the plane-wave-based real-time evolution[42,68,69]. In our calculations, the Kohn–Sham wavefunction, the density, and the Hamiltonian were self-consistently evolved through the time-dependent equation:

$$i\hbar\frac{\partial}{\partial t}\psi_{n,\mathbf{k}}(\mathbf{r},t) = \left[\frac{1}{2m}\left(-i\hbar\nabla + \frac{e}{c}\mathbf{A}_{\text{ext}}(t)\right)^2 + \sum_\lambda v_{\text{atom}}(\mathbf{R}_\lambda(t)) + V_{\text{DFT}}[\rho(\mathbf{r},t)]\right]\psi_{n,\mathbf{k}}(\mathbf{r},t), \tag{1}$$

where $n$ and $\mathbf{k}$ denote the band index and the Bloch momentum vector, respectively. $\mathbf{A}_{\text{ext}}$ and $V_{\text{DFT}}$ indicate the time-dependent vector potential and DFT potential, respectively. The discretized time step for the time integration ($\triangle t$) was set to 2.415 as. In our calculations, the electric field was expressed using the velocity gauge of the vector potential via the relation $\mathbf{E}(t) = -\frac{1}{c}\partial\mathbf{A}_{\text{ext}}/\partial t$. The initial wavefunctions $[\psi_{n,\mathbf{k}}(t=0)]$ were obtained from the static-ground-state DFT calculations using the QUANTUM ESPRESSO package[68] with the PBE exchange–correlation functional[70]. Spin–orbit coupling was not considered in our rt-TDDFT calculation to treat the large cell of TMD nanotubes. Using the time-evolving Bloch wavefunctions, we evaluated the time profile of the current density as follows[70]:

$$\mathbf{J}(t) = -\frac{e}{m}\sum_n\sum_{\mathbf{k}} f_{n,\mathbf{k}}\left\langle\psi_{n,\mathbf{k}}(t)|\hat{\boldsymbol{\pi}}|\psi_{n,\mathbf{k}}(t)\right\rangle \tag{2}$$

where $n$ is the band index, $f_{n,\mathbf{k}}$ is the initial occupation of the Bloch state, $m$ is the mass of an electron, and the gauge-invariant mechanical momentum is defined as $\hat{\boldsymbol{\pi}} = \frac{m}{i\hbar}[\hat{\mathbf{r}},\hat{H}] = \hat{\mathbf{p}} + \frac{e}{c}\mathbf{A}_{\text{ext}}(t) + i\frac{m}{\hbar}[V_{\text{NL}},\hat{\mathbf{r}}]$. We considered two different polarized lights in our calculation: $\mathbf{E}_\pm(t) = \pm\mathbf{E}_0\sin(\omega t)$. $\mathbf{J}_\pm(t)$ refers to the calculated current under external field $\mathbf{E}(t)$. To extract the second-order response of the estimated current, we used the even-order currents $\mathbf{J}_{\text{even}}(t) = \frac{(\mathbf{J}_+(t)+\mathbf{J}_-(t))}{2}$. To mitigate the noisy rapid oscillations, the even-order real-time currents were averaged over time: $\mathbf{J}_{\text{avg}}(t) = \frac{1}{T}\int_0^T\mathbf{J}_{\text{even}}(\tau)d\tau$. The atomic motion under a time-dependent potential (i.e., Ehrenfest dynamics) was determined by the time-dependent Hellmann–Feynman force, which was calculated as follows[69]:

$$\mathbf{F}_\lambda(t) = M_\lambda\frac{d^2\mathbf{R}_\lambda(t)}{dt^2} = \sum_n\sum_{\mathbf{k}}\left\langle\psi_{n,\mathbf{k}}(t)\left|\frac{d\hat{H}_{n,\mathbf{k}}}{d\mathbf{R}_\lambda}\right|\psi_{n,\mathbf{k}}(t)\right\rangle \tag{3}$$

where $M_\lambda$ and $\mathbf{R}_\lambda(t)$ are the atomic mass of the $\lambda$th atom and the position of the $\lambda$th atom, respectively.

## Data availability

The data presented in the main text are provided in the Source Data file. Additional data necessary for extension of the present work can be available from the authors on request. Source data are provided with this paper.

## Code availability

The tight-binding code for computing the shift current spectra is partially available from the github: https://github.com/BSKim12/Shift_current.git.

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

## Acknowledgements

B.K. and N.P. were supported by the National Research Foundation of Korea (NRF) grant funded by the Korea government (MSIT) (No. NRF-2019R1A2C2089332). J.K. was supported by an NRF grant funded by the Korea government (MSIT) (No. 2020R1F1A1048143). This work was supported by the National Supercomputing Center with supercomputing resources including technical support (KSC-2020-CRE-0101).

## Author contributions

J.K. conceived the idea of this study, B.K. performed first-principles calculations. B.K., N.P., and J.K. analyzed the data and wrote the manuscript together.

## Competing interests

The authors declare no competing interests.
