## [Peer Review File · Nature Communications]

Giant bulk photovoltaic effect driven by the wall-to-wall charge shift in WS₂ nanotubesREVIEWER COMMENTS

Reviewer #1 (Remarks to the Author):

The authors reported that WS₂ nanotubes exhibit a giant shift current near the infrared region as the wall-to-wall charge shift constituted a key advantage of the TMD nanotube geometry. Time-dependent density functional theory was carried out to study nonlinear effect of a strong field and the nonadiabatic effect of atomic motion. This work is interesting and may provide a solid basis for a complete quantum mechanical understanding of the unique light-matter interaction hidden in the geometric characteristics. It should be publishable after a major revision.

1. Experimentally, MW-/SW-TMD-NT had been reported. For the Junus TMD NTs, there has no report although their 2D counterparts are available in experiments. So, the stability of Junus NTs have to be confirmed.

2. As the bandgap increases with the size, the optical absorption was reduced. Why was the optical absorption spectrum insensitive? Fig. S3 showed the clear change.

3. Under the irradiation, electron shall be excited from valence top to conduction band. How to control the the transitions between different d orbitals in reality? Theoretically, the efficiency should be low if the situation is true.

4. For DWNT, the band gap should be smaller than those of individual NTs if there is type-II band alignment, leading to the enhanced large photovoltaic effect in calculation. What is the interwall effect? Fig. 3d showed the transition between two NTs. How to achieve that if the spacing is large? Normally, the intertube distance is large because of the weak bonding. Similar issue to the bundle in Fig. S10.

5. For Junus NTs, the bandgap was greatly reduced, which should be the main contribution to the improved photovoltaic effect. What is the effect of hot electron?

6. The interwall effect should be carefully considered, as it may not be possible in practical cases. Experimental realization of the design should be difficult.

Reviewer #2 (Remarks to the Author):

What are the noteworthy results?

The authors show that WS₂ nanotubes exhibit a large shift current near the infrared region, which is four times higher than the previously reported values in the higher frequency range. The authors state that the wall-to-wall charge shift constitutes a key advantage of the transition metal dichalcogenide (TMD) nanotube geometry. Moreover, the authors investigate a Janus-type heteroatomic configuration that maximizes the photovoltaic effect.

Their findings promote a quantum mechanical understanding of the unique light-matter interaction hidden in the geometric characteristics of the reduced dimension of the TMD materials.

Will the work be of significance to the field and related fields?

Yes

How does it compare to the established literature?

The report attempts to elucidate and understand the strong photovoltaic response found by Zhang, Y. J. et al. (Nature 570, 349-353 (2019)). The overall flow of the manuscript is very good and clear. I think that the findings presented by the author will promote the understanding of the unique light-matter interaction hidden in the geometric characteristics of the reduced dimension and the previously observed photovoltaic effect. Moreover, I believe their findings will serve as a guiding principle for designing and using the TMD nanotubes in advanced photovoltaic devices.

However, the authors fail to compare their work to the previous findings done by Prof. Seifert and Prof. Tenne, who pioneered this field. The results presented by the authors have to be compared to previous work so the novelty will stand out. Furthermore, substantial advances were achieved some

time ago on the topics of atomic structure and spontaneous polarization of SWNTs and electronic and optoelectronic properties of the zigzag SWNT.

To clarify, the author report very important and fundamental findings however, as they fail to refer to previous work, it is hard to differentiate between the authors' contribution and prior findings made by others.

I suggest addressing the following works in the introduction and for comparison to the reported results:

1. Seifert, G., Terrones, H., Terrones, M., Jungnickel, G. and Frauenheim, T., 2000. On the electronic structure of WS₂ nanotubes. Solid-state communications, 114(5), pp.245-248.

2. Seifert, G.; Terrones, H.; Terrones, M.; Jungnickel, G.; Frauenheim, T. 2000, Structure and electronic properties of MoS₂ nanotubes. Phys. Rev. Lett., 85 (1), 146.

3. Damnjanović, M., Vuković, T. and Milošević, I., 2017. Symmetry-based study of MoS₂ and WS₂ Nanotubes. Israel Journal of Chemistry, 57(6), pp.450-460.

4. Milošević, I., Nikolić, B., Dobardžić, E., Damnjanović, M., Popov, I. and Seifert, G., 2007. Electronic properties and optical spectra of MoS₂ and WS₂ nanotubes. Physical Review B, 76(23), p.233414.

5. Xia, H., Chen, X., Luo, S., Qin, F., Idelevich, A., Ghosh, S., Ideue, T., Iwasa, Y., Zak, A., Tenne, R., and Chen, Z., 2021. Probing the Chiral Domains and Excitonic States in Individual WS₂ Tubes by Second-Harmonic Generation. Nano Letters.

6. Milošević, I., Seifert, G., Popov, I., Dobardžić, E., Nikolić, B. and Damnjanović, M., 2006, September. Optical absorption in molybdenum disulfide nanotubes. In Nanomodeling II (Vol. 6328, p. 63280B). International Society for Optics and Photonics.

7. Zhang, D.B., Dumitrică, T. and Seifert, G., 2010. Helical nanotube structures of MoS₂ with intrinsic twisting: an objective molecular dynamics study. Physical review letters, 104(6), p.065502.

Does the work support the conclusions and claims, or is additional evidence needed?

No additional evidence needed

However, some claims and assumptions made by the authors are not supported by references, nor compared to previous works done in the field. (Especially in the sections of atomic structure and spontaneous polarization of SWNTs and electronic and optoelectronic properties of the zigzag SWNT). More details regarding the missing comparisons are in the attached file.

Are there any flaws in the data analysis, interpretation, and conclusions?

There are no serious flaws.

However, the authors fail to acknowledge similar or previous works done in this field. (see previous notes and attached file). Moreover, many findings are not discussed but rather stated. I think that authors should add additional discussion explaining the importance and implications of the results. (The details are in the attached file.)

Do these prohibit the publication or require revision?

Requires revision

Is the methodology sound?

Yes

Does the work meet the expected standards in your field?

Yes

However, there are no proper references to the previous work.

Is there enough detail provided in the methods for the work to be reproduced?

Yes

Specific Notes

Line 54: "...ferroelectric materials." – please add reference that supports this claim

Line 55: "... intertwined in this phenomenon," – please add reference that supports this claim

Line 56: "...inherent to the 1D structure." – please address previous works done on this topic

Line 56: "...d-d transitions of W atoms," – please address previous works done on this topic

Line 91-95: "...removing the ferroelectricity... zigzag... estimated polarization..." – please address previous works done on this topic, add reference that supports the claims.

Line 91-95: "...TMD monolayer." - please address previous works done on this topic, add reference that supports the claims.

Line 119-120: "...zone center (Γ point)" - please address previous works done on this topic, add reference that supports the claims. (For instance: Seifert, G. et al, 2007. Electronic properties and optical spectra of MoS₂ and WS₂ nanotubes. PRB, 76(23), p.233414.)

Line 132-135: "...because of the reduced optical..." - This point needs further clarification.

Line 142-148: "...We find that, rather than..." - please address previous works done on this topic, add reference that supports the claims.

Line 152-154: "...large negative shift-current peak." What is the importance of this finding? Please add discussion.

Line 226-229: "...value of the shift vector..." What are the implications of this finding? Please explain the importance of a larger shift vector.

Line 304-310: "We find that the..." - What is the importance of this finding? Also, there is no discussion on the following questions: (1) Why these frequencies were chosen? (2) Why do the chosen frequencies give the most interesting results?

Line 337-338: "As an ideal structure..." – There is no substantial discussion that support the claim that the Janus NT is the ideal system to probe wall to wall charge shift.

Reviewer #3 (Remarks to the Author):

The manuscript presents a systematic first-principles study on bulk photovoltaic effect (BPVE) of

transition-metal dichalcogenides (TMD) nanotubes. The calculation shows that the shift current is substantially enhanced in these one-dimensional nanotubes. Beyond single-wall nanotubes, the authors find that the shift currents of multi-wall nanotubes are enhanced by orders of magnitude because of the transitions between interwall states. They also predict that the Janus nanotubes can further enhance the shift-current photoconductivity by lowering the symmetry. In addition to the perturbation approach, the authors employ the time-dependent density functional theory to study the dynamics of structures under strong light illumination. They predict that the photocurrent can be switched by the intensity and frequency of the incident light. The results are interesting to ignite more experimental efforts to search for BPVE in one-dimensional nanostructures.

On the other hand, I have a few questions for the calculation and results.

1. The manuscript addresses that the interwall mechanism is important for the enhanced BPVE in multi-wall nanotubes. In addition to the (7,0)/(18,0) structure, they mentioned that this effect is also observed in bundles of single-wall, and triple-wall and other structures (figures S10-12). Those figures in the supplementary document do present some effects of interwall transitions. However, I cannot see the enhancement of BPVE. Compared with the figure 3c in the manuscript, those enhancements in supplementary figures are nearly negligible. Therefore, I have to question if the proposed giant BPVE and inter-wall transition mechanism in multi-wall nanotubes are robust.

2. I have a technical question. Because of the supercell structure and artificial vacuum in first-principles simulations, the VASP calculated dielectric function depends on the supercell size. Thus, the absolute value of the long-wave limit dielectric function and photoconductivity is not meaningful. Usually, we have to find an appropriate way to normalize them when compared with bulk results. Can the author explain how they eliminate this artificial factor in their results?

3. The authors claim that their double wall result agrees better with the measurements (Figure S9). However, I cannot see this conclusion, given that there are only three experimental data points. Particularly, density functional theory (DFT) is known for obtaining incorrect band gaps and lacking excitonic effects. Usually, the shape of DFT spectra is not reliable in one-dimensional structures. It is inappropriate to compare those detail features in high-energy regime. Moreover, Figure S9 does not specify the source of the experimental data. I assume that they are from Ref. 35. Unfortunately, Figure 1 (c) in Ref. 35 shows a much larger multiwall structures with a diameter of tens of nanometers. It is inappropriate to claim the agreement between results of the calculated small-double layer tubes and those "huge" nanotubes.

4. Finally, it is inappropriate to claim ferroelectricity in these nanotubes because they are just polar structures. Ferroelectricity needs the condition to switch the polarization by external field, which seems beyond the scope of this work.

We thank greatly all the reviewers for their pertinent highly valuable comments. We revised our manuscript in full accordance with the reviewers' comments, and here we provide our point-by-point responses to each of them, by quoting the original comment.

Reply to Reviewer #1

The authors reported that WS₂ nanotubes exhibits a giant shift current near the infrared region as the wall-to-wall charge shift constituted a key advantage of the TMD nanotube geometry. Time-dependent density functional theory was carried out to study nonlinear effect of a strong field and the nonadiabatic effect of atomic motion. This work is interesting and may provide a solid basis for a complete quantum mechanical understanding of the unique light–matter interaction hidden in the geometric characteristics. It should be publishable after a major revision.

We are grateful for the reviewer's encouraging comments and constructive suggestions. Below, we respond to each of the reviewer's comments.

1. Experimentally, MW-/SW-TMD-NT had been reported. For the Janus TMD NTs, there has no report although their 2D counterparts are available in experiments. So, the stability of Janus NTs have to be confirmed.

Figure R1-1. The first-principles molecular dynamics simulations of the zigzag (10,0) Janus-type WS₂e nanotube. **a** Variation of the temperature in the molecular dynamics simulation of the Janus-type nanotube. **b** Radial distribution function (RDF) of the Janus nanotube at 0.1 and 2 ps. **c** Snapshot of cross-sectional view of the geometry at 0, 1 and 2 ps.

As the reviewer pointed out, the practical synthesis of the Janus-type TMD NTs has not yet been reported, and we agree that the feasibility of the materials needs to be discussed more critically in terms of the stability. We tested the dynamical stability of Janus-type WS₂e nanotube by performing the Born-Oppenheimer *ab-initio* molecular dynamics. We considered a large superlattice of the single nanotube containing 180 atoms and examined whether the atomic configuration is robust during the dynamics near 300 K. As shown in Fig. R1-1, we confirmed the structural integrity is well maintained even near the thermal equilibrium reached at 2 ps. We find that a couple of previous articles also confirmed the stability of Janus-type NTs [*Phys. Chem. Chem. Phys.*, **20**, 20812-20820 (2018); *J. Mater. Chem. A*, **7**, 7885–7890 (2019)]. The article is cited in our revised text. (page 10, second paragraph line 5)

2. As the bandgap increases with the size, the optical absorption was reduced. Why was the optical absorption spectrum insensitive? Fig. S3 showed the clear change.

On the reviewer's comment, we now recognize that our expression '...insensitive to the size of the tube...' is somewhat misleading. Since the light absorption is inversely proportional to the energy difference between the initial and the final states, as the reviewer pointed out, the optical absorption

decreases with the tube size, as presented in Fig. S3a-d. In our previous manuscript, we intend to state that, despite this marginal variation of the optical absorption spectra, the overall spectral distribution of the absorption is preserved irrespective of the tube size (Fig. S3d). In the revised manuscript, we clearly state that the optical absorption decreases depending on the tube size. (page 6, first paragraph lines 6-8)

3. Under the irradiation, electron shall be excited from valence top to conduction band. How to control the transitions between different d orbitals in reality? Theoretically, the efficiency should be low if the situation is true.

Figure R1-2. Imaginary parts of the dielectric constants of Si bulk (Si), the (13,0) single-walled WS₂ bundle structure (TMD NT). The d orbital contribution of the imaginary dielectric constant of TMD NT (d - d transition) is represented by blue dashed line.

The transition rate between d orbitals should be very low for a single atom because optical selection rules dictates that the angular momentum changes are allowed only at $\Delta l = \pm 1$. Unlike the isolated atoms in spherical symmetric potentials, the selection rule for periodic crystals is not that straightforward; it depends on various physical quantities such as symmetry, atomic composition, magnetic ordering, and extrinsic parameters including substrate and defects. For example, the d - d transitions are explicitly demonstrated in well-known real materials, such as MoS₂ [*Nat. Commun.*, **3**, 887 (2012)]. The optical characteristics of TMD nanotubes are mostly inherited from monolayer TMD, and the lesson from the monolayer is very instructive. The optical absorption of WS₂ nanotube is not confined to the band edge transition but occurs in a wide energy window. Therefore, as demonstrated in our result, the optical absorption between d orbitals is sufficiently large to generate the giant shift current. For an explicit comparison, we show that the absorption strength of TMD nanotube is comparable to that of bulk Si (Fig. R1-2).

4. For DWNT, the band gap should be smaller than those of individual NTs if there is type-II band alignment, leading to the enhanced large photovoltaic effect in calculation. What is the interwall effect? Fig. 3d showed the transition between two NTs. How to achieve that if the spacing is large? Normally, the intertube distance is large because of the weak bonding. Similar issue to the bubble in Fig. S10.

Figure R1-3. The interwall effect depending on the interlayer distance. **a** Calculated shift-current spectra of coaxially aligned (10,0)@(30,0), (13,0)@(30,0), and (18,0)@(30,0) DWNTs with respect to various light frequencies. **b** Maximum peak of shift-current spectra of (10,0)@(30,0), (13,0)@(30,0), and (18,0)@(30,0) DWNTs at 2.3 eV with respect to the inter-tube distance. The inter-tube distance (d) is defined by the distance between W atoms in the outer and the inner tubes, as depicted in the inset of Fig. R1-3b.

We now recognize that our description of the band alignment of the DWNT can be inappropriately interpreted. The band alignment between the constituent tube is presented in Fig. 3a for the case of the (7,0)@(18,0) DWNT. Other configurations of DWNTs showed a similar trend. This band alignment can be classified as type-I, and the bandgap of the DWNT is determined by the inner tube. Shortly, we would like to emphasize that the enhancement of the shift current of the DWNTs is not attributable to the bandgap reduction. Instead, it originates from the interwall charge transfer on the excitation. As shown in Fig. 3a and Fig. 3c in the manuscript, the large peak of the shift current spectrum for the DWNT appears at 1.2 eV (downward arrow), while the bandgap of the DWNT is only 0.2 eV. This obviously indicates the enhanced shift current of the DWNT is not due to the band-edge excitations. On the reviewer's comment, we agree that our text need to be written in a clearer fashion, and we improved the discussion on the band alignment in the revised text. (page 9, line 4)

We definitely agree with the reviewer's concern that the interwall distance can be a critical factor governing the interwall charge shift. In response to the reviewer's question, we investigated the shift current of coaxial configurations of DWNTs with various interwall distances (Fig. R1-3). Although the shift current of DWNTs rapidly decreases with the increasing interwall distance, the interwall effect is

still larger than the intra-wall effect until the inter-W distance reaches 11 Å. In the revised manuscript, we added a brief description on the electronic origin and the limitation of the interwall effect. (page 9, lines 11-14)

5. For Janus NTs, the bandgap was greatly reduced, which should be the main contribution to the improved photovoltaic effect. What is the effect of hot electron?

The band structure of the (10,0) WS₂ SWNT and the same Janus-type SWNT are presented in Fig. 2 and Fig. S13, respectively. As the reviewer accurately pointed out, the bandgap of the Janus-type nanotube (0.11 eV) is substantially smaller than that of the WS₂ nanotube (0.51 eV). Although the bandgap is one of the main components affecting the dielectric constant, the overall magnitude of the dielectric constant of the Janus-type nanotube is mostly comparable to that of the WS₂ nanotube (Fig. 4b). On the other hand, the shift vector of the Janus-type nanotube is distinctly different from that of the WS₂ nanotube (Fig. 4c). This sharply enhanced shift vector provides a primary source for the enhanced shift current of the Janus-type nanotube, as indicated by the downward arrow in Fig. 4a. Shortly, in this case, the advantage of smaller bandgap is only marginal, and the main advantage of the Janus-type SWNT should be attributed to the low-lying peak of the shift vector (Fig. 4c).

The shift current considered in the present work is a combined effect of two intrinsic photo-responses of a semiconductor system: one is the absorption strength, and the other is the shift vector on the excitation. In our computation of these intrinsic properties, carrier dynamics or scattering processes are not included, and the surplus kinetic energy of electrons, created by high-energy photons or high-voltage electrodes, are not considered. To understand more realistic photo-responses of the carrier dynamics, many extrinsic factors need to be considered, including the effect of scattering and relaxation, which is far more extensive, beyond the scope of the present study.

6. The interwall effect should be carefully considered, as it may not be possible in practical cases. Experimental realization of the design should be difficult.

We strongly agree with the reviewer's point. For an explicit assessment of the interwall effect, the optical measurement of the SWNT samples and the DWNTs sample need to be compared, which is hard to implement in practical experiment. The large photovoltaic characteristics observed in an experiment cannot be explained by the *ab-initio* calculations results of the SWNT, and the synthesized nanotube samples are mostly reported as multi-walled nanotubes [*Nature*, **570**, 349 (2019)]. Here we

suggest a possibility that the interwall effect is an essential ingredient for the photovoltaic effect. To relieve the concern of the reviewer and convey more accurate information, we specified the limitations and implications of our theoretical approach and the necessary conditions for the giant shift current in the revised manuscript. (page 9, lines 11-14)

We believe that we responded to every detail of the questions raised by the reviewer and revised the manuscript accordingly. In the revised version, we added the dynamical stability of Janus NTs and modified some misleading parts of the text. We clarified the concept of the interwall effect and clearly stated the physical implication and the limitation of our results. Therefore, we believe that the revised manuscript is now suitable for publication in *Nature Communications*.

Reply to Reviewer #2

The authors show that WS₂ nanotubes exhibit a large shift current near the infrared region, which is four times higher than the previously reported values in the higher frequency range. The authors state that the wall-to-wall charge shift constitutes a key advantage of the transition metal dichalcogenide (TMD) nanotube geometry. Moreover, the authors investigate a Janus-type heteroatomic configuration that maximizes the photovoltaic effect.

Their findings promote a quantum mechanical understanding of the unique light-matter interaction hidden in the geometric characteristics of the reduced dimension of the TMD materials.

We are very grateful for the reviewer's careful reading and positive assessment of our findings. Below we present our answers for each of the questions.

1. The report attempts to elucidate and understand the strong photovoltaic response found by Zhang, Y. J. et al. (Nature 570, 349-353 (2019)). The overall flow of the manuscript is very good and clear.

I think that the findings presented by the author will promote the understanding of the unique light-matter interaction hidden in the geometric characteristics of the reduced dimension and the previously observed photovoltaic effect. Moreover, I believe their findings will serve as a guiding principle for designing and using the TMD nanotubes in advanced photovoltaic devices.

However, the authors fail to compare their work to the previous findings done by Prof. Seifert and Prof. Tenne, who pioneered this field. The results presented by the authors have to be compared to previous work so the novelty will stand out. Furthermore, substantial advances were achieved some time ago on the topics of atomic structure and spontaneous polarization of SWNTs and electronic and optoelectronic properties of the zigzag SWNT.

To clarify, the author report very important and fundamental findings however, as they fail to refer to previous work, it is hard to differentiate between the authors' contribution and prior findings made by others.

I suggest addressing the following works in the introduction and for comparison to the reported results:

- (1) Seifert, G., Terrones, H., Terrones, M., Jungnickel, G. and Frauenheim, T., 2000. On the electronic structure of WS₂ nanotubes. *Solid-state communications*, 114(5), pp.245-248.
- (2) Seifert, G.; Terrones, H.; Terrones, M.; Jungnickel, G.; Frauenheim, T. 2000, Structure and electronic properties of MoS₂ nanotubes. *Phys. Rev. Lett.*, 85 (1), 146.
- (3) Damnjanović, M., Vuković, T. and Milošević, I., 2017. Symmetry-based study of MoS₂ and WS₂ Nanotubes. *Israel Journal of Chemistry*, 57(6), pp.450-460.
- (4) Milošević, I., Nikolić, B., Dobardžić, E., Damnjanović, M., Popov, I. and Seifert, G., 2007. Electronic properties and optical spectra of MoS₂ and WS₂ nanotubes. *Physical Review B*, 76(23), p.233414.
- (5) Xia, H., Chen, X., Luo, S., Qin, F., Iidevich, A., Ghosh, S., Ideue, T., Iwasa, Y., Zak, A., Tenne, R., and Chen, Z., 2021. Probing the Chiral Domains and Excitonic States in Individual WS₂ Tubes by Second-Harmonic Generation. *Nano Letters*.
- (6) Milošević, I., Seifert, G., Popov, I., Dobardžić, E., Nikolić, B. and Damnjanović, M., 2006, September. Optical absorption in molybdenum disulfide nanotubes. In *Nanomodeling II* (Vol. 6328, p. 63280B). International Society for Optics and Photonics.
- (7) Zhang, D.B., Dumitrică, T. and Seifert, G., 2010. Helical nanotube structures of MoS₂ with intrinsic twisting: an objective molecular dynamics study. *Physical review letters*, 104(6), p.065502.

We appreciate greatly that the reviewer brings to our attention the previous articles. In the revision, we introduce and cite those suggested references, and clarify the advances made by our present work compared with the previous works. (page 4, first paragraph line 4; page 5, first paragraph line 3; page 5, first paragraph line 10; page 6, first paragraph line 4)

2. - Line 54: "...ferroelectric materials." – please add reference that supports this claim.

- Line 55: "... intertwined in this phenomenon," – please add reference that supports this claim.

- Line 56: "...inherent to the 1D structure." – please address previous works done on this topic.

- Line 56: "...d-d transitions of W atoms," – please address previous works done on this topic.

- Line 91-95: "...removing the ferroelectricity... zigzag... estimated polarization..." – please address previous works done on this topic, add reference that supports the claims.

- Line 91-95: "...TMD monolayer." - please address previous works done on this topic, add reference

that supports the claims.

- Line 119-120: "...zone center (Γ point)" - please address previous works done on this topic, add reference that supports the claims. (For instance: Seifert, G. et al, 2007. Electronic properties and optical spectra of MoS₂ and WS₂ nanotubes. PRB, 76(23), p.233414.).

- Line 142-148: "...We find that, rather than..." - please address previous works done on this topic, add reference that supports the claims.

We are grateful for these very detailed suggestions. In this revision we cited the suggested articles in accordance with the reviewer's comments. (page 3, first paragraph lines 1-4; page 3, second paragraph line 6; page 4, first paragraph line 4; page 5, first paragraph line 3; page 5, first paragraph line 10; page 6, first paragraph line 4; page 7, second paragraph line 9)

3. Line 132-135: "Various ($n,0$) SWNTs exhibit large peak values of the shift current over a wide energy range; however, their maximum values somewhat decrease with increasing tube size because of the reduced optical absorption (Fig. S3)" - This point needs further clarification.

Optical absorption is inversely proportional to the energy difference between initial and final states. As we explained in the manuscript, the band gap increases with the size of SWNT and thus its optical absorption strength naturally decreases with the diameter of the tube. In Fig. S3, we present that the overall trend of the shift current can be associated with the variations in the absorption strength, which depends on the bandgap determined by the tube diameter. In the revised manuscript, we rephrase the sentence in question more clearly and added the references related to our findings. (page 6, last two sentences of first paragraph)

4. Line 152-154: "Under the higher-frequency light radiation ($\hbar\omega = 2.2$ eV), the positive SDOS is strongly suppressed, leading to a large negative shift-current peak." What is the importance of this finding? Please add discussion.

The shift current spectrum of Fig. 2c has different signs depending on the light frequency. To analyze this sign change we introduced the SDOS which is presented in Fig. 2d and defined in our previous work (Ref. [40]). The positive contribution decreases with increasing frequency whereas the negative contribution is dominant at high frequency. By tuning the applied light, we can manipulate both the direction and magnitude of the generated photocurrent. Therefore, the SDOS is very useful to

understand the sign change of the shift current depending on the driving light frequency. Following the reviewer's suggestion, we added a brief discussion on the implication of the SDOS result. (page 7, the last line of second paragraph)

5. Line 226-229: "Whereas the optical absorption does not substantially differ from that of the WS₂ DWNT (Fig. 4e), the value of the shift vector for the Janus DWNTs is more than three times greater than that for the pristine DWNTs (Fig. 4f)." What are the implications of this finding? Please explain the importance of a larger shift vector.

Shift current is determined by the product of shift vector and optical absorption [*Phys. Rev. B* **61**, 5337 (2000)]. Optical absorption provides information on how many electrons are excited by an external light. Shift vector is the charge-center difference between initial and final states. A large shift vector indicates that, on the excitation, the electrons travel a long distance in the position space. The optical absorption of the Janus DWNT is comparable to that of the WS₂ DWNT (Fig. 4e). However, the shift vector of Janus DWNT is greatly enhanced (Fig. 4f), owing to the presence of the radial asymmetric potential. In short, the advantage of Janus DWNT is not from the bandgap or absorption strength, but from the shift vector, intensified by the interwall effect. In this revision, we added an explanation on the definition of shift vector (page 11, first paragraph lines 5-6) and the physical mechanism underlies the large shift vector of the Janus DWNT (page 11, second paragraph lines 6-7)

6. Line 304-310: "We find that the exposure to light with a frequency of $\hbar\omega = 0.9$ eV leads to the spontaneous displacement of outer sulfur atoms along the radial direction (inset of Fig. 5f), which greatly enhances the generated photocurrent." - What is the importance of this finding? Also, there is no discussion on the following questions: (1) Why these frequencies were chosen? (2) Why do the chosen frequencies give the most interesting results?

(1) As shown in the inset of Fig. 5a, the shift current spectrum of the zigzag (7,0) WS₂ exhibits peaks, of which the low-lying peaks are at 0.9 eV and 1.7 eV. For $\hbar\omega=1.7$ eV, the excited state carriers are dispersed over a wide range of energies, while the hole and electron carriers are located sharply near the band edges when excited with $\hbar\omega=0.9$ eV (Fig. 5b). In the main text, we choose to present mainly the latter case as the time variation of the carrier distribution is more apparently traceable. As presented in Fig. 5g-i, the substantial effect of atomic motion on the photocurrent is apparently visualized in terms of the variation of the electron density. However, the effect of atomic motions is not necessarily unique to the chosen frequency ($\hbar\omega=0.9$ eV), a similar effect with $\hbar\omega=1.7$ eV is given below.

(2) The physical significance of this effect of atomic motions is discussed with Fig. 5h. As stated in the manuscript, the excited electrons occupy W-S antibonding states (Fig. 5h), leading to elongation of W-S bonding. This atomic motion triggers additional orbital mixings and, as a result, the electron/hole carriers are distributed in a wider energy/real space (Fig. 5h and 5i). This interesting behavior is not limited to the light with frequency $\hbar\omega=0.9$ eV. We carried out the same calculations with $\hbar\omega=1.7$ eV and find that the atomic motion leads to an enhancement of the photocurrent. In the revised manuscript, by inserting the results for $\hbar\omega=1.7$ eV (Fig. S20), we elaborated the discussion that this effect of atomic movement can be observed for other frequencies as well. (page 16, the last lines of the first paragraph)

7. Line 337-338: “As an ideal structure than can maximize the wall-to-wall charge shift, we considered Janus-type nanotubes.” – There is no substantial discussion that support the claim that the Janus NT is the ideal system to probe wall to wall charge shift.

On the reviewer’s comment, we recognize that it is too strong to state that the Janus-type DWNT is “an ideal structure” for the enhanced wall-to-wall charge shift. However, the comparison of the shift vectors between WS₂ DWNT and Janus-type DWNT (Fig. 4f) obviously indicates that the heterogeneous chemical composition of Janus-type DWNT contributes to the amplification of the shift vector. To relieve the reviewer’s concern and to deliver our message more clearly, we modified the text accordingly. (page 16, second paragraph lines 6-7)

We again appreciate the reviewer’s consideration of our manuscript as a potential candidate for publication in *Nature Communications* and his/her valuable suggestions reinforcing the manuscript. We believe the raised issues are all settled, and we are convinced that the revised manuscript is ready for publication in *Nature Communications*.

Reply to Reviewer #3

The manuscript presents a systematic first-principles study on bulk photovoltaic effect (BPVE) of transition-metal dichalcogenides (TMD) nanotubes. The calculation shows that the shift current is substantially enhanced in these one-dimensional nanotubes. Beyond single-wall nanotubes, the authors find that the shift currents of multi-wall nanotubes are enhanced by orders of magnitude because of the transitions between interwall states. They also predict that the Janus nanotubes can further enhance the shift-current photoconductivity by lowering the symmetry. In addition to the perturbation approach, the authors employ the time-dependent density functional theory to study the dynamics of structures under strong light illumination. They predict that the photocurrent can be switched by the intensity and frequency of the incident light. The results are interesting to ignite more experimental efforts to search for BPVE in one-dimensional nanostructures.

We greatly appreciate the reviewer's thoughtful reading and constructive suggestions. Below, we provide our point-by-point responses to each of the reviewer's comment.

On the other hand, I have a few questions for the calculation and results.

1. The manuscript addresses that the interwall mechanism is important for the enhanced BPVE in multi-wall nanotubes. In addition to the (7,0)/(18,0) structure, they mentioned that this effect is also observed in bundles of single-wall, and triple-wall and other structures (figures S10-12). Those figures in the supplementary document do present some effects of interwall transitions. However, I cannot see the enhancement of BPVE. Compared with the figure 3c in the manuscript, those enhancements in supplementary figures are nearly negligible. Therefore, I have to question if the proposed giant BPVE and inter-wall transition mechanism in multi-wall nanotubes are robust.

Figure R3-1. The shift current spectra of the (13,0) SWNTs packed in a hexagonal lattice. Two different in-plane lattice parameters are compared in these computations to test the dependence of the charge shift on the wall-to-wall distance.

On the reviewer's comment, we now recognize that our presentation of the interwall effect for the cases of bundles and multi-walled nanotubes is not sufficient. In Fig. R3-1, we tested the shift current for the SWNTs bundles, closely packed in a hexagonal lattice. We tested with two different in-plane lattice parameters to quantify the effect of the wall-to-wall distance. We obviously observe that the shift current of the SWNT bundle is enhanced over that of isolated SWNT. The shift current enhancement is also sensitive to the inter-tube distance because the increase is induced by the inter-wall effect. However, such enhancement of the shift current in this bundle configuration is not so dramatic compared with that of DWNT (see Fig. 3c in the main text and Fig. R3-3 in this letter). This difference can obviously be attributed to the fact that the contact area between the two walls of the interfacing tubes, in this bundle case, is substantially smaller than that of the coaxial geometry.

Figure R3-2. Projected density of states (PDOS) of the triple-walled nanotube (TWNT) without spin-orbit coupling. The inset indicate top view of the TWNT.

The reviewer also requested more convincing evidence of the interwall effect in multi-walled structures. In Fig. S11, we presented the dielectric constant and the shift current spectra for the DWNTs and the triple-walled nanotube. Compared with the SWNT, the enhancement induced by the interwall effect is sharply appreciable in the case of DWNT. However, the enhancement of the triple-walled nanotube, compared with the DWNT, is very marginal in the low energy region ($< 2\text{eV}$). The effect of triple walls is only noticeable in the higher energy region ($> 2\text{eV}$). These are direct consequences of the band alignment. As shown in Fig. R3-2, the band gap increases with the size of the SWNT, and the coaxial multi-walled tubes constitute the type-I band alignment, as explained in Fig. 3a in the main text. With this type-I band alignment, the interwall effect is led by the inner most two walls for the low energy region, while it will be dominated by the outer two walls as the light frequency increases. Figure S11c features such dependence of the multi-walls on the light frequency. In summary, although the multi-walled ones bring the same interwall effect in the charge shift, because of the diameter-dependent

energy band configurations, the interwall effect between two pairs of coaxial tubes are manifested in different regions of the energy window.

Figure R3-3. The interwall effect depending on the interlayer distance. **a** Calculated shift-current spectra of coaxially aligned (10,0)@(30,0), (13,0)@(30,0), and (18,0)@(30,0) DWNTs with respect to various light frequencies. **b** Maximum peak of shift-current spectra of (10,0)@(30,0), (13,0)@(30,0), and (18,0)@(30,0) DWNTs at 2.3 eV with respect to the inter-tube distance. The inter-tube distance (d) is defined by the distance between W atoms of the outer and the inner tubes, depicted in the inset of Fig. R3-3b.

Last but not least, we would like to emphasize that the charge transfer between the walls of the interfacing tubes is responsible for the large photovoltaic effect. As above mentioned, the interwall effect decreases as the separation between the walls increases. However, as summarized in Fig. R3-3, the interwall effect is still larger than the intra-wall effect until the interwall distance (the distance between W atoms, as denoted in the inset of Fig. R3-3) reaches 11 Å, which indicates the interwall effect is efficient and robust even in a large cell. In the revised manuscript, we strengthened the descriptions of the interwall effect in multi-walled tubes and bundles. The data presented in this letter (Fig. R3-3) are summarized in the supplementary information Fig. S9. (page 9, first paragraph lines 11-14)

2. I have a technical question. Because of the supercell structure and artificial vacuum in first-principles simulations, the VASP calculated dielectric function depends on the supercell size. Thus, the absolute value of the long-wave limit dielectric function and photoconductivity is not meaningful. Usually, we have to find an appropriate way to normalize them when compared with bulk results. Can the author explain how they eliminate this artificial factor in their results?

Figure R3-4. Imaginary parts of the dielectric constants of the isolated (10,0) WS₂ SWNT calculated with two different vacuum thickness.

We fully agree with the reviewer that unwanted artifacts can be introduced when we simulate a low-dimensional system in a supercell structure with vacuum. (1) To avoid the use of ill-defined physical quantities in low dimensions, in our calculations, we focus on the axial components (z -direction) of optical/optoelectronic properties, i.e., well-defined responses of the one-dimensional periodic direction. (2) To eliminate the ambiguity induced by the non-periodicity in the plane direction (xy -direction), as shown in Fig. R3-4, we normalize the optical/optoelectrical quantities, considering the presence of the vacuum region, normal to the axial direction [*npj 2D Mater. Appl.* **2**, 6 (2018); *Nat. Commun.* **12**, 4330 (2021)]. While the bulk quantity is normalized by the volume of the unit cell, the dielectric constant and the shift current spectrum in this 1D case are normalized by the unit length in the axial direction. As a result, the dielectric constant is independent of the choice of the vacuum thickness in the plane direction, and the unit of the shift current is changed from A/V^2 to $\text{A} \text{ \AA}^2/\text{V}^2$. In Fig. R3-4, we explicitly present that such renormalized dielectric constant is indeed independent of the selected thickness of the vacuum. We definitely agree with the reviewer’s concern, and we added a brief explanation on the change of the unit in consideration of the vacuum layer in the Methods section. (page 17, second part of method section lines 5-9)

3. The authors claim that their double wall result agrees better with the measurements (Figure S9). However, I cannot see this conclusion, given that there are only three experimental data points. Particularly, density functional theory (DFT) is known for obtaining incorrect band gaps and lacking excitonic effects. Usually, the shape of DFT spectra is not reliable in one-dimensional structures. It is inappropriate to compare those detail features in high-energy regime. Moreover, Figure S9 does not specify the source of the experimental data. I assume that they are from Ref. 35. Unfortunately, Figure 1 (c) in Ref. 35 shows a much larger multiwall structures with a diameter of tens of nanometers. It is inappropriate to claim the agreement between results of the calculated small-double layer tubes and those “huge” nanotubes.

We thank the reviewer for bringing an important issue to our attention. We agree with the reviewer that the direct matching our result with the experiment is not appropriate given that the experimental data are not sufficient and also the DFT approach has technical limitations. As the reviewer pointed out, the DFT scheme does not rigorously deal with the quasi-particle energy spectra, and the electron-hole two-body interaction is absent. Following the reviewer's suggestion, we removed the sentence in question to convey a more accurate information to readers. We also specified the sources of all experimental data in the revised text.

However, our main finding is the enhanced charge shift owing to the inter-wall effect, and we think the inter-wall effect in the multi-walls can be generically described with the effect in the double walls. As we replied to the first question of the reviewer, the inter-wall effect is not cumulative in the same energy region for the multi-walls. Instead, as the band alignments of the multi-walls are all type-I, the inter-wall effects of each interfacing wall are manifested in a different region of the energy window: the inter-wall effect of the inner tubes appears in low energy region, while that of outer tubes arises in higher energy region. Shortly, the primary mechanism of the inter-wall charge shift in the multi-walls is mostly identical to that of double walls. Nevertheless, we agree with the reviewer's concern that unnecessarily overinterpretation must be avoided, and we added a brief discussion on the validity of the inter-wall effect in a more realistic situation instead of claiming the direct agreement with the experiment. (page 9, lines 9-11)

4. Finally, it is inappropriate to claim ferroelectricity in these nanotubes because they are just polar structures. Ferroelectricity needs the condition to switch the polarization by external field, which seems beyond the scope of this work.

We are very grateful for this knowledgeable comment of the reviewer. In this revision, we changed the word "ferroelectricity" to "electric polarization".

We believe that we answered all the questions raised by the reviewer and revised the manuscript appropriately. In the revised version, the validity of interwall effect is justified in a larger DWNT, and the somewhat inappropriate expressions are modified/removed to strengthen our arguments. Therefore, we believe that the revised manuscript is now suitable for publication in *Nature Communications*.

REVIEWER COMMENTS

Reviewer #1 (Remarks to the Author):

The revision is publishable.

Reviewer #2 (Remarks to the Author):

What are the noteworthy results?

This paper deals with the shift current of zigzag WS₂ nanotubes, which is the source of the large bulk photovoltaic effect (BPVE) in such nanotubes, as demonstrated experimentally in 2019. That is the first attempt to calculate the BPVE in such nanotubes from the first principle theory. The authors use perturbation theory in the framework of the time-dependent Schrodinger equation to derive the BPVE. To my understanding, the authors do not apply an external bias on the nanotube in order to avoid competition with the usual photocurrent of such nanotubes from bandgap excitation. Obviously, though, there is no power out without a bias-dependent photocurrent, which is obtained in regular PV cells by an external load. The beauty of the paper lies in its ability to grasp the salient contributions to the BPVE in such nanotubes using the high-level first-principle calculations. In the experiments, however, the nanotubes are appreciable larger, which leads to effects not considered by the authors of this MS. If I properly understand the calculations, the lifetime of the free carriers does not play any role, hence, the influence of realistic effects, like defect density and scattering, does not play any role here.

The paper is of high quality and is suitable for publication

I liked the fact that the excited electron and hole live in different layers. However, it is important to note that it is analogous to type II excitons in heterolayers, which provides means for a long lifetime of the excited carriers.

The last part of the work (non-adiabatic BPVE) is a true "tour de force", and I liked it very much.

That is a very important and timely paper, and I recommend it for publication.

Will the work be of significance to the field and related fields?

Yes

How does it compare to the established literature?

Adequate

Does the work support the conclusions and claims, or is additional evidence needed?

No additional evidence is needed.

Here are several more comments:

I have no way to compare the BPVE currents produced here to those obtained experimentally in WS₂ bulk under AM1.5 solar radiation, i.e., 100 mW/cm² (about 20-25 mA/cm²). I wish to understand if the BPVE is on the same order of magnitude, which I suspect very much. Also, I guess the authors did not do the calculations under bias, but this is very important to know if power can be extracted from the nanotubes.

I know there is no way the authors can calculate the BPVE for real-size nanotubes, which are 30-100 nm in size and contain 10-20 layers. Smaller nanotubes cannot be reliably synthesized presently. But there is a huge difference between their calculations and the reality (Nature 2019) for several reasons: 1. Quantum size and strain effects are relevant to very small nanotubes, as presented here (< 3 nm in diameter), and are much smaller for the large nanotubes for which the BPVE was measured. 2. Consequently, the IR response, which is highly relevant for the present calculations, is not likely to exist in the normal size nanotubes. 3. The effect of chirality is enormous in real-size nanotubes with pitches (unit cell size) is probably 20 nm in the axial direction. The large diameter leads to enormous asymmetry in the nanotube. Moreover, each layer is probably of a different chirality, and most of the layers are indirect bandgap materials. Thus, I believe that most of the shift current stems from the chirality, rather than the zigzag configuration, as it is calculated here.

I do not think the authors have to recalculate their data, but they should discuss these points briefly in

the text and give proper citations.

Are there any flaws in the data analysis, interpretation, and conclusions?

There are no serious flaws.

Do these prohibit the publication or require revision?

No additional revision required

Is the methodology sound?

Yes

Does the work meet the expected standards in your field?

Yes.

Is there enough detail provided in the methods for the work to be reproduced?

Yes

Reviewer #3 (Remarks to the Author):

The reply and revised manuscript well answered my questions and comments. I recommend it to be published.

One optional comment is about the discussion of the energy details and, particularly, the shift current direction of the highly-energy range on page 7. It is known that excitonic effects will dramatically modify the optical spectra. So the discussion based on DFT DOS is not reliable. It is expected that the shift current direction may be changed for different-energy photons. However, the quantitative discussion on page 7 is not necessary.

Reply to Reviewer #1

The revision is publishable.

We greatly appreciate the reviewer for the recommendation of the publication of our manuscript.

Reply to Reviewer #2

This paper deals with the shift current of zigzag WS₂ nanotubes, which is the source of the large bulk photovoltaic effect (BPVE) in such nanotubes, as demonstrated experimentally in 2019. That is the first attempt to calculate the BPVE in such nanotubes from the first principle theory. The authors use perturbation theory in the framework of the time-dependent Schrodinger equation to derive the BPVE. To my understanding, the authors do not apply an external bias on the nanotube in order to avoid competition with the usual photocurrent of such nanotubes from bandgap excitation. Obviously, though, there is no power out without a bias-dependent photocurrent, which is obtained in regular PV cells by an external load. The beauty of the paper lies in its ability to grasp the salient contributions to the BPVE in such nanotubes using the high-level first-principle calculations. In the experiments, however, the nanotubes are appreciable larger, which leads to effects not considered by the authors of this MS. If I properly understand the calculations, the lifetime of the free carriers does not play any role, hence, the influence of realistic effects, like defect density and scattering, does not play any role here.

The paper is of high quality and is suitable for publication

I liked the fact that the excited electron and hole live in different layers. However, it is important to note that it is analogous to type II excitons in heterolayers, which provides means for a long lifetime of the excited carriers.

The last part of the work (non-adiabatic BPVE) is a true “tour de force”, and I liked it very much.

That is a very important and timely paper, and I recommend it for publication.

We greatly appreciate for the reviewer’s strong recommendation for the publication and very positive assessment on the timeliness of our manuscript. The reviewer exactly catches our scheme of the BPVE which focuses on the intrinsic key components of the shift current mechanism, independent of the external bias. Using the eigenstate spectra of the Kohn-Sham equation of the density functional theory, we evaluate the response function of photo-induced current based on perturbation theory. This computation scheme is suited to the calculation of the intrinsic material properties and hard to include the effect of external loads which is incommensurate with the periodic unit-cell potential.

I have no way to compare the BPVE currents produced here to those obtained experimentally in WS₂ bulk under AM1.5 solar radiation, i.e., 100 mW/cm² (about 20-25 mA/cm²). I wish to understand if the BPVE is on the same order of magnitude, which I suspect very much. Also, I guess the authors did

not do the calculations under bias, but this is very important to know if power can be extracted from the nanotubes.

Figure R2-1. The calculated photocurrents of double-walled WS₂ nanotubes in the present work and the measured ones in the previous experiment (Fig. 3d in our manuscript ref. 35) with the multi-walled samples.

We focus on the dominant intrinsic properties of the WS₂ materials for the large BPVE, while various sources of extrinsic effects, such as lifetime of carriers and effects of scatters, are not considered in our calculations. As the reviewer pointed out, the quantitative comparison with the experimental results can be a touchstone for validation of our approach. For comparison, we explicitly present computed and observed photocurrents of one-dimensional WS₂ nanotubes in Fig. R2-1. We converted our theoretical values by considering experimental conditions (longitudinal length of crystal: 5 μm and impedance of WS₂ nanotube: $6 \times 10^7 \Omega$) for three different wavelengths (532, 632.8, 730 nm) with 10^2 W/cm² laser intensity. It is greatly noteworthy that, despite the great disparity in the tube size between our models (double walls) and the experiment (thick multiwalls), the amounts of photocurrents are largely comparable. This agreement obviously indicates not only the dominance of the intrinsic mechanism, over various extrinsic effects such as scattering and carrier lifetime, but more importantly the validity of our double-walled model. Below we discuss that, as our calculations of the double-walled tubes captures the essence of the interwall charge shift, it serves as generic model for triple-walled tubes or further thick multi-walled tubes.

I know there is no way the authors can calculate the BPVE for real-size nanotubes, which are 30-100 nm in size and contain 10-20 layers. Smaller nanotubes cannot be reliably synthesized presently. But there is a huge difference between their calculations and the reality (Nature 2019) for several reasons: 1. Quantum size and strain effects are relevant to very small nanotubes, as presented here (< 3 nm in diameter), and are much smaller for the large nanotubes for which the BPVE was measured. 2.

Consequently, the IR response, which is highly relevant for the present calculations, is not likely to exist in the normal size nanotubes.

Figure R2-2. The effect of the interwall charge shift of DWNTs with various interlayer distances. a Calculated shift-current spectra of coaxially aligned (10,0)@(30,0), (13,0)@(30,0), and (18,0)@(30,0) DWNTs with respect to various light frequencies. **b** Maximum peak of shift-current spectra of (10,0)@(30,0), (13,0)@(30,0), and (18,0)@(30,0) DWNTs at 2.3 eV with respect to the inter-tube distance. The inter-tube distance (d) is defined by the distance between W atoms of the outer and the inner tubes, depicted in the inset of Fig. R2-2b.

We appreciate for the reviewer's very pertinent comment and the exact appreciation of the scope of our study. The main interest of this study is the intrinsic photo-responsivity of the nanotube, and we showed that the interwall charge shift is an essential component for the large shift current in double-walled nanotube; the mechanism is also thought to be applicable for the large-size nanotubes. We first emphasize that this intriguing interwall charge shift is distinct from the advantage of the dimensionality lowering (from 2D to 1D nanotube) which was highlighted in ref. 35. The effect of the interwall charge shift between coaxial or adjacent tubes provides a new photo-active conducting channel, leading to the dramatic enhancement of photocurrent over those of monolayers or single-walled tubes. It is noteworthy that this interwall charge shifting mechanism is quite robustly persistent over large wall-to-wall separations, as presented Fig. R2-2.

Figure R2-3. Comparison of the optoelectronic properties of the DWNT and that of the triple-walled nanotube. **a** Calculated band structure of a (7,0)@(18,0)@(32,0) WS₂ triple-walled nanotube (TWNT) without spin-orbit coupling calculation. **b** Imaginary parts of the dielectric constants of the DWNT and the TWNT without spin-orbit coupling calculation. The inset is top view of the TWNT. **c** Calculated shift current spectra of the DWNT and the TWNT with respect to the frequency of the applied light without spin-orbit coupling.

Figure R2-4 Projected density of states (PDOS) of the triple-walled nanotube (TWNT) without spin-orbit coupling. The inset depicts the cross-sectional view of the TWNT geometry.

The interwall effect in thick multi-walled tubes can be deduced from the comparison between the DWNTs and the triple-walled nanotubes (TWNTs), as summarized in Fig. R2-3. Note that, compared with the SWNT, the DWNT exhibits sharply enhanced the shift current (see Fig. 3 in the main text). However, the enhancement by the triple walls, compared with the DWNT, is very marginal in the low energy region ($< 2\text{eV}$); the effect of triple walls is appreciable only in the higher energy region ($> 2\text{eV}$), as presented in Fig. R2-3b and Fig. R2-3c. These are apparent consequences of the band alignment. As shown in Fig. R2-4, the band gap increases with the size of the tubes, and thus the interwall effect in

the low energy region is governed by the pairs of walls in inner shells, while the effect in higher energy ranges is dominated by the outer pairs of walls. Overall, in this regard, the results of the double-walled tube feature the essence of the interwall effect even in thick multi-walled ones at a given energy window.

We appreciate for the pertinence of the reviewer's comment and, in the revised version, we added a brief discussion on the implication of the interwall charge shift mechanism in realistic thick multi-walled nanotubes (page 10, second paragraph lines 1-14).

3. The effect of chirality is enormous in real-size nanotubes with pitches (unit cell size) is probably 20 nm in the axial direction. The large diameter leads to enormous asymmetry in the nanotube. Moreover, each layer is probably of a different chirality, and most of the layers are indirect bandgap materials. Thus, I believe that most of the shift current stems from the chirality, rather than the zigzag configuration, as it is calculated here.

I do not think the authors have to recalculate their data, but they should discuss these points briefly in the text and give proper citations.

We fully agree with the reviewer that realistic samples contain diverse chiral nanotubes which are possibly featured by indirect bandgaps. However, the shift current occurs via the direct excitations between the occupied and the unoccupied bands, as guided by the second-order formula [*Phys. Rev. B* **61**, 5337 (2000)]. Further, as shown in Fig. 3, the main optical absorption responsible for the large shift current is not driven by the band edge excitation and, for the interwall charge shift in WS₂ nanotubes, the energy levels of *d* orbitals is likely to be more important than the band dispersion near the band edges. Therefore, our conclusion drawn by calculations with the zigzag nanotube would still serve as a leading mechanism for the interwall charge shift in multi-walled nanotubes with various chiralities. We added a brief discussion on possible effects of the chiral nanotubes on the interwall charge shift in the revised manuscript (page 10, second paragraph lines 14-20).

Reply to Reviewer #3

The reply and revised manuscript well answered my questions and comments. I recommend it to be published.

We thank the reviewer for careful reading of our manuscript and the recommendation of publication.

One optional comment is about the discussion of the energy details and, particularly, the shift current direction of the highly-energy range on page 7. It is known that excitonic effects will dramatically modify the optical spectra. So the discussion based on DFT DOS is not reliable. It is expected that the shift current direction may be changed for different-energy photons. However, the quantitative discussion on page 7 is not necessary.

As the reviewer mentioned, the excitonic effects can be manifested by apparent optical absorption peaks near band-edge transitions [*J. Appl. Phys.* **122**, 084301 (2017); *Sci. Rep.* **11**, 7683 (2021)]. However, the large shift currents, the main focus of our present study, are usually driven by higher energy absorptions than the bandgap excitations as featured by the shift current peaks in 1.0, 1.5, and 2.2 eV in Fig. 2d. Moreover, the main practical purpose of the BPVE is usually aimed at the utilization of solar energy harvesting at room temperature, for which the excitonic effect is suppressed rapidly. It is noteworthy that the calculated shift currents without considering the excitonic effect successfully reproduce experimental results [*Phys. Rev. Lett.* **109**, 116601 (2012)]. The direction of the shift current direction is solely determined by the shift vector, which is rarely affected by the excitonic effect [*PNAS* **118**, e1906938118 (2021)], and the excitonic effect is present mainly at the band edge absorption strength in low temperature. Therefore, although the details of the SDOS can be somewhat adjusted by the excitonic effect, the analysis on the shift current generation mechanism, particularly for the interwall charge shift mechanism, is still valid irrespective of the limitation of the practical Kohn-Sham treatment. We added a brief discussion about the excitonic effect on the shift current in the revised version of the manuscript (page 18, lines 10-15).

REVIEWERS' COMMENTS

Reviewer #2 (Remarks to the Author):

The authors addressed all my questions and comments in their answers and the revised manuscript. The paper is of high quality, and I recommend it be published

Reply to Reviewer #2

The authors addressed all my questions and comments in their answers and the revised manuscript.

The paper is of high quality, and I recommend it be published.

We greatly appreciate the reviewer for the acknowledging the value of our works and recommendation of the publication of our manuscript.